# No Sex Differences in the Attentional Bias for the Right Side of Human Bodies

Chiara Lucafò [1,2,*], Daniele Marzoli [1,*], Cosimo Ferrara [1], Maurizio Bertollo [2] and Luca Tommasi [1]

1   Department of Psychological Sciences, Health and Territorial, University of Chieti, Via dei Vestini 29, 66100 Chieti, Italy
2   Department of Medicine and Aging Sciences, Behavioral Imaging and Neural Dynamics (BIND) Center, University "G. d'Annunzio" of Chieti-Pescara, Via dei Vestini 29, 66100 Chieti, Italy
*   Correspondence: chiara.lucafo@unich.it (C.L.); d.marzoli@unich.it (D.M.)

**Abstract:** Ambiguous silhouettes representing human individuals which perform unimanual actions are interpreted more often as right-handed. Such a preference might reflect a perceptual frequency effect, due to the fact that most social interactions occur with right-handers. As a consequence, observers would preferentially attend to the region in which others' dominant hand usually falls, thus increasing the efficiency in monitoring both aggressive and communicative acts. Given that men can be more dangerous compared with women, the right-hand bias should be larger when observing male rather than female individuals, and given that aggressive interactions involve men more frequently than women, it should be larger in male rather than female observers. However, previous studies did not specifically test whether: (i) male—compared with female—observers pay more attention to the right side of others (regardless of the observed individuals' sex), or (ii) observers (regardless of their sex) pay more attention to the right side of male—compared with female—individuals. Therefore, in the present study we used ambiguous human silhouettes rotating about their vertical axis with one arm extended in order to determine whether the rightward bias is larger for male rather than female figures and/or in male rather than female participants. According to our data, the bias toward the right side of human bodies was not significantly associated with either the figure's or the participant's sex.

**Keywords:** handedness; sex; human body; ambiguous figures; perceptual frequency effect





## 1. Introduction

Several studies indicate a perceptual and attentional bias toward the right side of others' bodies in both right- and left-handers. For example, people tend to imagine right-handed individuals [1–4], and to perceive ambiguous silhouettes representing human individuals which perform unimanual actions as right-limbed [5–11]. Moreover, a perceptual frequency effect [12] has been proposed to explain the advantage that left-handed people exhibit in interactive sports, namely those which involve a physical contest with one or more opponents [13–15]. For instance, an advantage of left-handedness has been reported for various combat sports [16], including boxing [17–19], karate and taekwondo [20], wrestling [21], mixed martial arts [19,22,23], and fencing [24,25]. Consistent findings have also been observed in non-combat sports such as basketball [26], baseball [27], cricket [28], and tennis [29–35]. Some authors accounted for the left-handers' advantage in sports with the so-called innate superiority hypothesis, which claims that left-handers are endowed with greater motor, spatial, and visual abilities compared with right-handers. However, other authors reject such an explanation and suggest some kind of tactical advantage (see [7] for a more detailed discussion). In particular, a common explanation is that the benefit of left-handed players is due to their opponents having little experience with playing against left-handed individuals, namely the negative frequency-dependent selection hypothesis [12]. Indeed, given that left-handers represent only about 10% of the general population [36,37], people usually play against right-handers, with the consequence that

left-handers could be particularly advantaged in interactive sports because they utilize unfamiliar playing strategies and patterns of attack [14,38]. This could cause some problems because the required motor responses are underpracticed, with the consequence that the defensive reactions would be less automatic and effective [30]. Similarly, usual offensive strategies may fail against left-handers (e.g., as regards tennis, the backhand is generally less effective than the forehand, so that, when playing against a left-handed opponent, a player should accustom themselves to this uncommon situation [33]). As already proposed [1–7,9–11,39,40], the bias to perceive right-handed actions might be a result of the preference to attend to the region in which others' dominant hand usually falls. Whereas such a tendency could entail an increased efficiency in monitoring both aggressive and communicative acts, given that the right rather than the left limb is preferred for both types of behavior, the drawback would be a reduced monitoring of the region in which the left hand falls. Thus, the left-handers' advantage in fighting [12] and sports (for a review, see [15]) might arise from a reduced ability in the discrimination of left- rather than right-handed movements. The present study is theoretically grounded in the negative frequency-dependent selection hypothesis rather than in the innate superiority hypothesis, and we believe that the bias to perceive ambiguous human silhouettes as right-limbed observed in previous studies [5–11] can be accounted for by the frequent interaction with right-handed and right-footed individuals. It should be remarked that this interpretation is in agreement with the finding that the visual presentation of right- and left-handed actions intensifies and attenuates, respectively, the higher predictability of the outcome of right- rather than left-handed actions [41].

In previous research using ambiguous stimuli, we reported some findings suggestive of a stronger preference for perceiving right-limbed movements in male rather than female observers ([6], Experiment 1 in [7]). From an evolutionary perspective, such findings might be accounted for by the higher incidence of physical fights among male rather than female individuals [42], which could have resulted in a larger bias toward the right side of men rather than women, as well as in a greater advantage of male left-handers compared with female left-handers in interactive sports. In line with this reasoning, tennis research suggests that the left-handers' advantage might be modulated by sex, because it seems to be larger for male compared with female players [15,29,34,43–45]. For example, Breznik [29] found that male left-handed professional tennis players are more successful in competitions than their female counterparts. In this regard, the male bias observed in several studies—according to which point-light walkers are perceived more frequently as male than as female [46–48]—is not astonishing, because misinterpreting a man for a woman can be more costly than misinterpreting a woman for a man, with men usually representing more threatening stimuli compared with women. The facing bias reported for human biological motion (according to which point-light walkers without explicit depth cues are interpreted more frequently as facing toward the viewer than as facing away from the viewer; [46,49], see also [5,7,8]) seems also to be linked to the perceived sex of the figure. Indeed, such a bias is larger when the figure is perceived as male rather than female [46,50], and even larger in male observers than in female observers [46]. In summary, given that men might be more dangerous compared with women, the right-hand bias should be larger when observing male rather than female individuals, and given that aggressive interactions involve men more frequently than women, it should be larger in male rather than female observers. Such hypotheses would be in agreement with the aforementioned finding that the advantage of left-handed players is larger for males rather than females [15,29,34,43–45]. However, since previous research has examined males playing against males and females playing against females, they do not allow the discrimination of whether male observers (compared with female observers) pay more attention to the right side of others (regardless of the observed individuals' sex), or whether observers (regardless of their sex) pay more attention to the right side of male rather than female individuals. On the other hand, testing both female and male participants with both female and male stimuli would allow the discrimination between the two possible explanations. In line with these considerations,

the aim of our study was to demonstrate whether the rightward bias is larger for male rather than female silhouettes and/or in male rather than female participants.

## 2. Materials and Method

Following the recommendations proposed by Simmons, Nelson, and Simonsohn [51], we provide detailed information on how we defined the sample size, as well as on manipulations, measures, and data exclusions carried out in the present study.

### 2.1. Participants

Sixty-four participants (32 females and 32 males; age: 18–30 years) took part in the study. All participants had normal or corrected-to-normal vision. For each combination of response arrow spinning direction (clockwise, CW, or counterclockwise, CCW), color (red or green), and position (above or below; see Procedure section), we scheduled the recruitment of eight female participants and eight male participants.

### 2.2. Stimuli

As stimuli, we used 256 animations obtained from two original animations, created with the software Poser Pro 2012 (Smith Micro Software Inc., Pittsburgh, PA, USA), one representing the silhouette of a human female and the other of a human male (the two silhouettes visually differed by their physical appearance, and more specifically by their sexually dimorphic traits, such as different waist-to-hip ratios and the presence of breasts in the female one). The silhouette stood on its legs with one arm extended (Figure 1) and rotated about its vertical axis while maintaining a static posture. The silhouette was depicted in black against a white background and no clear-cut depth cue was available, so that the animation was ambiguous and could be perceived as rotating either CW or CCW. The two original animations were decomposed into their 32 constituent frames using Adobe Photoshop CS6 (Adobe, San Jose, CA, USA). Then, we created 64 different versions for each animation. Each version consisted of a whole spin of the silhouette, and was obtained by rearranging the 32 frames on the basis of starting frame and order (i.e., from the 1st frame to the 32nd and vice versa, from the 2nd frame to the 1st and vice versa, and so on). This order manipulation counterbalanced the association between extended arm and spinning direction, given that in the original order (which can be defined as "palmward" with reference to the hand movement) the CW rotation was congruent with an extended left arm, whereas in the inverted order (which can be defined as "backward" with reference to the hand movement) the CW rotation was congruent with an extended right arm, and vice versa for the CCW rotation. Although the Poser parameters (such as camera elevation and distance) were set so as to remove—as far as possible—potential perspective cues (such as relative height and relative size), a second set of 128 animations was created by mirroring, horizontally, each frame. This manipulation counterbalanced the effects of any remaining uncontrolled depth cue on the extended arm. Therefore, the final set of stimuli consisted of 256 animations, which represented each possible combination of the figure's sex (female or male), starting frame, type of rotation (palmward or backward movement), and mirroring (the whole set of frames constituting the different stimuli is available in the Open Science Framework repository at: https://osf.io/djx5v, accessed on 7 February 2023). The component frames of each stimulus measured around 10.1° vertically and, on average, around 3.7° horizontally at a viewing distance of 57 cm.

### 2.3. Procedure

The experiment was run using SuperLab 4.0 (Cedrus Corporation, San Pedro, CA, USA) on a Windows (Microsoft Corporation, Redmond, WA, USA) notebook with an Intel (Intel Corporation, Santa Clara, CA, USA) processor and a 15.6-inch monitor. Participants, seated comfortably in a quiet room with their eyes about 57 cm from the computer screen, were instructed to place their hands palm-down on the table and not to cross their legs, arms, or even fingers throughout the experiment, which consisted of two 128-trial blocks,

one with the female silhouette and the other with the male silhouette as stimuli. After the first block, participants were allowed to take a short break before the second block (the order of block presentation was counterbalanced across participants). In each trial, a black fixation cross, presented for 500 ms in the center of a white screen, was followed by one of the previously described stimuli, presented centrally for 2300 ms, and then by a pair of colored arrows, one slightly above and one slightly below the center of the screen, which represented the two possible spinning directions of the silhouette (see Figure 2). Participants were required to gaze at the fixation point and to indicate the perceived spinning direction of the silhouette by responding with the word "ROSSO" (the Italian word for "RED") or "VERDE" ("GREEN") on the basis of the arrow corresponding to their percept. The experimenter, seated behind the participant, recorded the participant's response by pressing the key "R" or "V" on a keyboard connected to the computer, and then the next trial started. In each block, stimuli were shown in a random sequence.

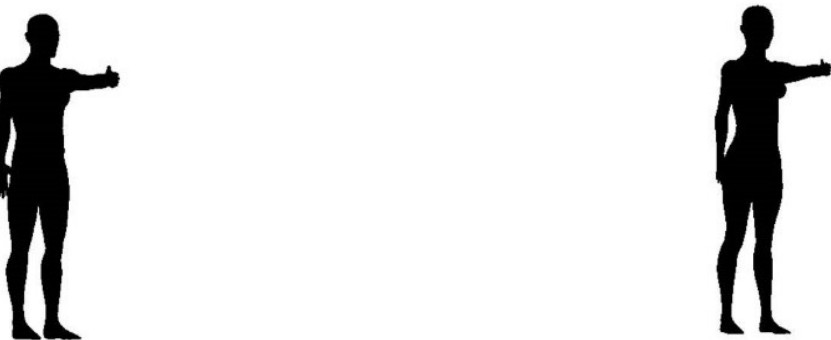

**Figure 1.** Examples of male (**left**) and female (**right**) stimuli.

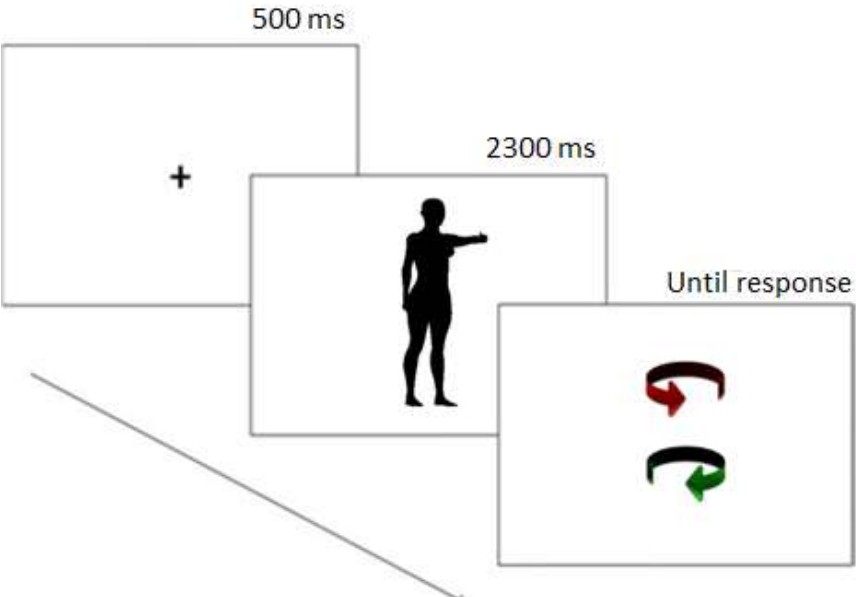

**Figure 2.** Examples of timeline.

The starting frame of each stimulus lasted 750 ms, whereas each of the following 31 frames lasted 50 ms (this stratagem aimed to reduce the possible carry-over of responses from trial to trial; e.g., see [52]). Moreover, we instructed the participants to provide their responses by selecting one of the two colored arrows, which represented the two spinning directions (CW or CCW), rather than by asking them to provide simple vocal responses, because labeling as CW or CCW a rotation about an axis approximately parallel to their own body axis can be rather difficult for participants. Before the experiment, participants were familiarized with the response modality by administering them a pretest in which they

used the two response arrows to indicate the spinning direction of a black human silhouette which included a clear-cut perspective clue, such as the relative size of the hands in different positions. This stratagem allowed the exclusion from the study of any individual unable to carry out the task. Finally, participants' hand preference was assessed by administering them the Italian version of the Edinburgh Handedness Inventory [53]. The study was conducted according to the guidelines of the Declaration of Helsinki, and approved by the Institutional Review Board of Psychology of the Department of Psychological Sciences, Health and Territory of the University of Chieti (protocol code 21003). All participants provided written informed consent.

*2.4. Data Analysis*

Fifty-eight participants (with a positive laterality score on the Italian version of the Edinburgh Handedness Inventory [53] [range: 0.15/1.00; M = 0.65 ± 0.09 SEM]) were classified as right-handers, and six participants (with a negative laterality score [range: 0.80/0.19; M = 0.54 ± 0.28 SEM]) were classified as left-handers.

We aimed to examine whether the rightward bias for male stimuli overcomes the rightward bias for female stimuli. First of all, we discarded from the data analysis two male subjects that gave the same response—'ROSSO' or 'VERDE'—to all the 256 trials. Then, we disregarded six female participants and four male participants who scored more than two standard deviations above or below the mean of their 'Sex' group in the number of figures perceived as right-handed for any combination of the figure's sex (female or male) and type of rotation (palmward or backward). After testing whether the number of figures perceived as right-handed was larger than the number of figures perceived as left-handed, we performed a repeated-measures analysis of variance (ANOVA) on the number of figures perceived as right-handed, using type of rotation (palmward or backward) and figure's sex (female or male) as within-subjects factors, and participant's sex (female or male) as the between-subjects factor. Because of the limited number of left-handers, we did not include handedness as an independent variable in the analysis of variance performed, but we correlated the laterality score with the number of figures perceived as right-handed.

## 3. Results

The number of figures perceived as right-handed (M = 139.54 [54.51%]) was larger than the number of figures perceived as left-handed (M = 116.46 [45.49%]; $t_{51}$ = 5.54; $p < 0.001$). The ANOVA showed no significant effect (Table 1).

**Table 1.** ANOVA results and corresponding effect size estimates.

| Factors | df | F | Sig. | $\eta_p^2$ |
|---|---|---|---|---|
| participant's sex | 1,50 | 1.501 | 0.226 | 0.029 |
| figure's sex | 1,50 | 0.556 | 0.459 | 0.011 |
| figure's sex X participant's sex | 1,50 | 2.322 | 0.134 | 0.044 |
| type of rotation | 1,50 | 2.342 | 0.132 | 0.045 |
| type of rotation X participant's sex | 1,50 | 0.042 | 0.839 | 0.001 |
| figure's sex X type of rotation | 1,50 | 0.440 | 0.510 | 0.009 |
| figure's sex X type of rotation X participant's sex | 1,50 | 3.581 | 0.064 | 0.067 |

As shown in Figure 3, both sex groups perceived a significantly larger number of right-handed figures (female participants: $t_{25}$ = 4.60; $p < 0.001$; male participants: $t_{25}$ = 3.22; $p < 0.005$). As shown in Figure 4, participants perceived a significantly larger number of right-handed figures for both female and male stimuli (female stimuli: $t_{51}$ = 4.36; $p < 0.001$; male stimuli: $t_{51}$ = 5.26; $p < 0.001$). Moreover, no significant correlation was found between the number of figures perceived as right-handed and the laterality score ($r_{52}$ = −0.06; ns).

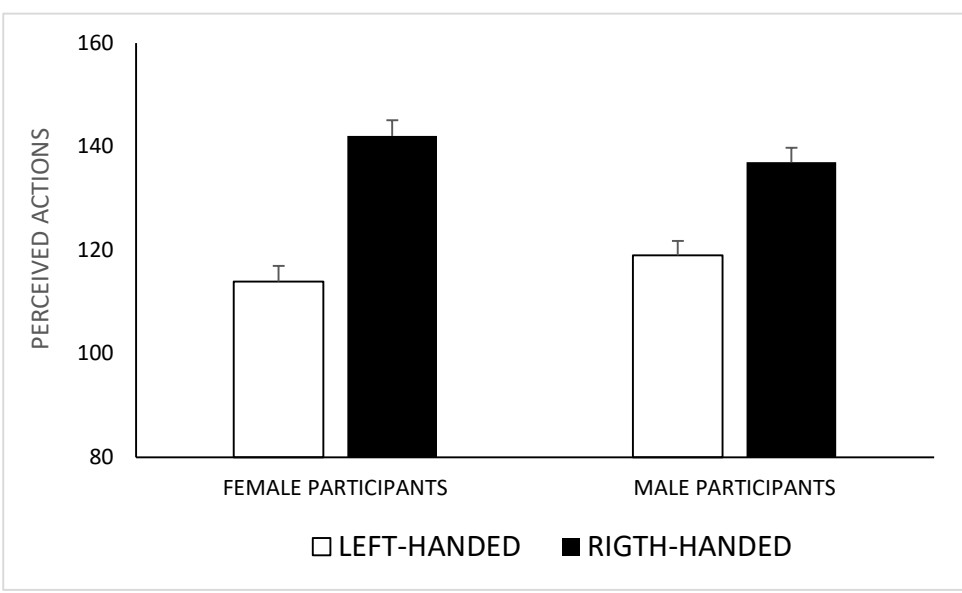

**Figure 3.** Mean number (and SEM) of actions perceived as right- and left-handed by female and male participants.

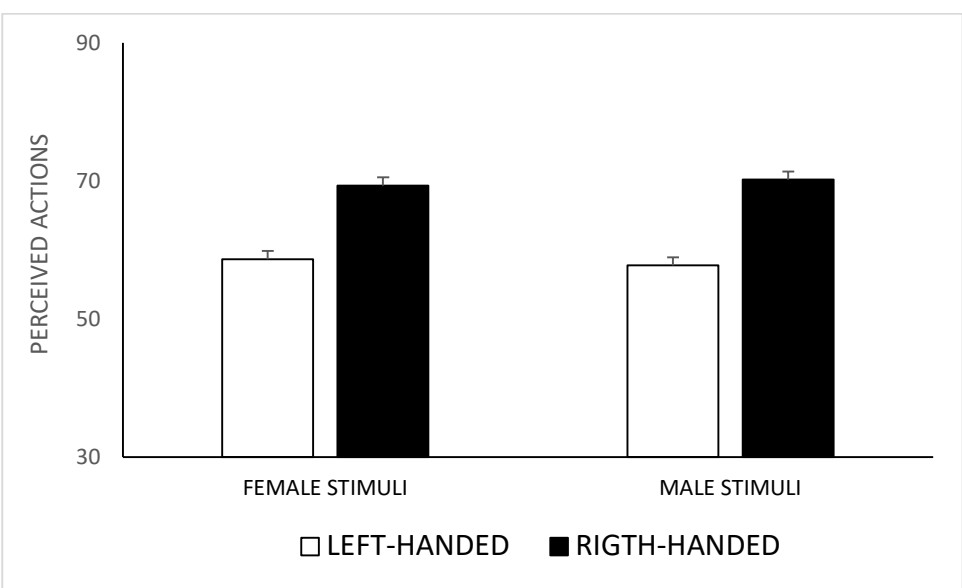

**Figure 4.** Mean number (and SEM) of actions perceived as right- and left-handed for female and male stimuli.

## 4. Discussion

The present study corroborated the existence of a bias toward the right side of observed bodies, as already shown in several previous studies [1–11]. Indeed, regardless of the figure's sex (female or male) and the participant's sex (female or male), participants showed a preference to perceive the silhouette spinning consistently with a right- rather than a left-hand action. Therefore, the predictions that the bias for the right hand would have been stronger (1) in the male rather than the female participants or (2) when observing male rather than female individuals were not confirmed. Actually, in our previous research with ambiguous stimuli we found only circumstantial clues to a stronger preference for perceiving right-limbed movements in male rather than female observers, and they might be instances of type I errors ([6], Experiment 1 in [7]). The present data are thus not in line with those from sport studies indicating that the left-handers' advantage is larger for male than for female players [15,29,34,43–45]. However, it should be noticed that the large

prevalence of right-handed actions in the real world (about 90%) is reflected in a slight prevalence of ambiguous human bodies perceived as right-limbed (in most cases, about 53%) [5–11]. Therefore, it is plausible that, if any sex difference exists in attentional and perceptual asymmetries toward the right side of others, it may not be strong enough to modulate accordingly the tendency to perceive ambiguous human silhouettes as right-handed, having to act on an extremely narrow range. Another possibility is that the relatively low level of interaction implied in the stimuli (i.e., an isolated individual rotating about their vertical axis with one arm extended) might have prevented the emergence of the hypothesized sex differences. On the other hand, the present results further support the generalizability of the bias to interpret ambiguous human bodies as right-handed rather than left-handed. In fact, whereas previous studies showed that the right-limb bias is observed both for static [8–11] and dynamic [5–7,10] stimuli, and for both hand [6–11] and foot [5–7] actions, the finding reported here demonstrates how the ubiquity of the bias extends to human bodies with different features, such as the sexually dimorphic traits of female and male bodies. However, other individual differences might modulate the bias toward the right side of human bodies. For instance, testing different age groups might determine whether such a bias is innate (in which case it should be comparable in younger and older individuals) or acquired (in which case it should increase with age, as occurs for asymmetries in face perception; for a review, see [54]). Moreover, it could be examined whether the bias toward the right side of human bodies is reduced in individuals with autistic traits. Indeed, it should be noticed that (1) individuals with autism exhibit impaired configural processing [55] and (2) inversion—which impairs the configural processing of human bodies (e.g., [56,57])—extinguished the preference to perceive right-handed actions in ambiguous human bodies [10]. Asymmetries in face perception also seem to be absent [58] or delayed [59] in individuals with autism. Therefore, an interesting question is whether the bias toward the right side of human bodies is related to perceptual asymmetries for faces, as well as to any factors affecting the latter (e.g., maternal cradling preferences, anxiety, emotional context; see [39] for a more detailed discussion). In this regard, we point out that the long-standing observation (e.g., [60,61]) of a perceptual bias in favor of the right side of human faces (which falls in the observer's left visual field) cannot be accounted for by asymmetric characteristics of the faces themselves, especially in light of the fact that the left hemiface is usually more expressive than the right hemiface (e.g., see [62] for a more detailed discussion). On the contrary, we have proposed [39,40] that the continuous interaction with right-handed individuals might foster the emergence of a rightward bias not only for bodies but also for faces (and hence the hypothesized associations). Future studies specifically addressing these issues are warranted.

**Author Contributions:** Conceptualization, C.L. and D.M.; methodology, C.L., C.F. and D.M.; formal analysis, C.L. and D.M.; investigation, C.L. and C.F.; resources, M.B. and L.T.; writing—original draft preparation, C.L. and D.M.; writing—review and editing, C.F., M.B. and L.T.; visualization, C.L.; supervision, D.M. and L.T.; project administration, D.M. All authors have read and agreed to the published version of the manuscript.

**Funding:** This research received no external funding.

**Institutional Review Board Statement:** The study was conducted according to the guidelines of the Declaration of Helsinki, and approved by the Institutional Review Board of Psychology of the Department of Psychological Sciences, Health and Territory of the University of Chieti (protocol code 21003).

**Informed Consent Statement:** Informed consent was obtained from all subjects involved in the study.

**Data Availability Statement:** All stimuli and datasets generated and analyzed during the current study are available in the Open Science Framework repository at: https://osf.io/36d7y/?view_only= (accessed on 7 February 2023).

**Conflicts of Interest:** The authors declare no conflict of interest.

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
