# Peer review of "No Sex Differences in the Attentional Bias for the Right Side of Human Bodies"

_symmetry, doi:10.3390/sym15020466_

Round 1
Reviewer 1 Report
It is interesting that the study tested both female and male participants with both female and male stimuli about whether the right bias is larger for male rather than female participants/stimuli.
1. Why use the spinning silhouette as the stimuli rather than other more interactively actions? The less interaction is a possible reason why the study did not find the expected result. Another possibility is that clockwise is more often than counterclockwise in our daily life.
2. Whether the position of the red arrow and the green arrow was counterbalanced? Is it another possible reason why the study did not find the expected result?
3. I would like to see the detailed results of ANOVA, especially whether there was an interaction between the figure’s sex and the participant’s sex.
4. Since the variable about left-handed and right-handed is participants’ judgment, perhaps it is better to treat it (e.g., the proportion of the right-handed judgment) as a dependent variable rather than an independent variable.
5. And how about the results of the right-handed participants? Similar with or different from the results from whole participants (including 6 left-handed participants)?
6. It would be clearer to have a figure of the trial procedure.
7. Please indicate the title of the Y-axis in Figure 3.
Reviewer 2 Report
My suggestions are:
1. Literature gap should be given.
2. Step by step explanation should be given.
3. More papers can be cited. For instance
- Ensemble residual network-based gender and activity recognition method with signals
- Gender Classification Using Gait Energy Images and One Shot Learning
- An automated daily sports activities and gender recognition method based on novel multikernel local diamond pattern using sensor signals
Or other suitable papers can be discussed. You are free to select appropriate papers.
4. Block references should not be given. References should be edited
5. The dataset used should be presented clearly and concisely in the table.
6. Conclusions section should be added.
Reviewer 3 Report
In the present study the authors investigated whether the bias for the right side of human bodies is modulated by the gender of the participants and the gender of the ambiguous stimuli used. The authors found that participants' gender and stimuli's gender do not seem to affect the bias
Overall, I find the topic of this manuscript interesting and I believe that the data collected has the potential to be of interest. However, there are some shortcomings that prevent the publication of the manuscript in the present form.
- Abstract: not sure if the link wit left-handed players is relevant here; I would delate this and add some more info about the results
- Introduction: I think that the Authors should explain in more details what the 'advantage' of left-handed people consists of (i.e., better accuracy or shorter reaction times?). Also, I find a little bit confusing the expression 'bias toward the right side of others' body', since in the tasks described the bias is really in judging whether an ambiguous body parts look like the right or the left hand, and not in observing more the right part of a body vs. the left one.
- Method: I find the beginning of section 1.1.1 obvious and not needed. Is there a difference in age between the female and male group? Will the author be able to provide more stimuli examples (the text description only could be a little confusing)? Why response was not given directly by the participants by keyboard press? Effect sizes (like the eta square) should be reported, and the stats for the non-significant interactions and correlations are needed too. Figures need to be improved, for example axis titles are missing. As for the stats I think it would be also important to include one-sample t-tests to verify that the number of stimuli judged as left- or right-handed are different from the chance level (which would be 128 for 256 stimuli - indeed if participants' responses are not different than chance than the main effect of laterality becomes difficulty interpretable).
- Discussions: I think that the Authors should include a more detailed discussion if the left-perceptual bias (LPB) for faces. Indeed, different studies (e.g. Wolfe, 1933; Gilbert & Bakan, 1973) have shown that people tend to spend more time observing the part of the face that falls within their left visual fields (hence the right part of the other people face). There might be some common mechanisms behind the LPB for faces and the right bias observed by the Authors.
Round 2
Reviewer 1 Report
The authors have made response to my comments. However, some of them still need further revision.
1. In Table 1, "*" should be replaced by "x". The "," under the columns of "F", "Sig.", and "ηp2" should be ".".
2. From Figures 2 and 3, I think the authors still treat type of rotation as an independent variable. I suggest to treat the percentage of right-handed judgment (or the percentage of left-handed judgment since their sum equal to 1) as the dependent variable, and treat figures' sex, participants' sex as independent variables to conduct ANOVA and draw a figure in which there is information about the percentage of right-handed judgment for each sex of partipants judge each sex of stimuli. And also to conduct single-sample t-tests for each condition with 50%. If the percentage right-handed judgment was significantly larger than 50%, then we say a right-handed bias was observed.
Reviewer 3 Report
The authors have replied to all my comments. Just one last point: shouldn't the df be different for main effects and interactions? I still think that the addition of one-sample t tests would be valuable.
Round 3
Reviewer 1 Report
I have no further comments or suggestions.